# Aptamers with Self-Loading Drug Payload and pH-Controlled Drug Release for Targeted Chemotherapy

**DOI:** 10.3390/pharmaceutics13081221

**Published:** 2021-08-07

**Authors:** Zihua Zeng, Jianjun Qi, Quanyuan Wan, Youli Zu

**Affiliations:** Department of Pathology and Genomic Medicine, Houston Methodist Hospital, 6565 Fannin Street, Houston, TX 77030, USA; zzeng@houstonmethodist.org (Z.Z.); JQi@houstonmethodist.org (J.Q.); qwan@houstonmethodist.org (Q.W.)

**Keywords:** aptamer–drug conjugate, doxorubicin, high drug payload, self-loading, targeted chemotherapy

## Abstract

Doxorubicin (DOX) is a common anti-tumor drug that binds to DNA or RNA via non-covalent intercalation between G-C sequences. As a therapeutic agent, DOX has been used to form aptamer–drug conjugates for targeted cancer therapy in vitro and in vivo. To improve the therapeutic potential of aptamer–DOX conjugates, we synthesized trifurcated Newkome-type monomer (TNM) structures with three DOX molecules bound through pH-sensitive hydrazone bonds to formulate TNM-DOX. The aptamer–TNM–DOX conjugate (Apt–TNM-DOX) was produced through a simple self-loading process. Chemical validation revealed that Apt–TNM-DOX stably carried high drug payloads of 15 DOX molecules per aptamer sequence. Functional characterization showed that DOX payload release from Apt–TNM-DOX was pH-dependent and occurred at pH 5.0, which reflects the microenvironment of tumor cell lysosomes. Further, Apt–TNM-DOX specifically targeted lymphoma cells without affecting off-target control cells. Aptamer-mediated cell binding resulted in the uptake of Apt–TNM-DOX into targeted cells and the release of DOX payload within cell lysosomes to inhibit growth of targeted lymphoma cells. The Apt–TNM-DOX provides a simple, non-toxic approach to develop aptamer-based targeted therapeutics and may reduce the non-specific side effects associated with traditional chemotherapy.

## 1. Introduction

Targeted antitumor drug delivery allows for drug accumulation within tumor sites via targeted delivery vehicles to improve drug efficacy and minimize side effects [1,2]. The target molecule, delivery vehicle, and antitumor drug are the main components of a targeted antitumor drug delivery system. Target molecules are often expressed on the tumor cell surface. A variety of target-specific ligands, including folate, transferrin, aptamers, antibodies, and peptides, have been investigated to guide targeted antitumor drug delivery [3].

Aptamers are short single-stranded DNA (ssDNA) or RNA molecules that can form unique 3D structures to bind their targets with high specificity and affinity. Cell-specific aptamers can be developed using a cell-Systematic Evolution of Ligands by Exponential enrichment (SELEX) procedure [4]. Using a hybrid SELEX strategy that combines cell-SELEX and protein-SELEX, we developed a ssDNA aptamer targeting CD30-expressing T cell lymphoma and classic Hodgkin lymphoma cells [5]. We further demonstrated the utility of this aptamer as a delivery vehicle for targeted chemotherapy and immunotherapy [6,7,8,9].

Antitumor drugs can be loaded into aptamer sequences via chemical synthesis for covalent conjugation or a simple non-covalent loading approach [10,11]. Doxorubicin (DOX), a chemotherapy drug used to treat solid tumors and hematologic malignancies [12], is of interest for aptamer-mediated drug delivery, as DOX can simply and non-covalently intercalate into GC regions of aptamer sequences [13,14]. However, GC region scarcity reduces aptamer DOX loading capacity, which restricts the effectiveness of aptamer-based targeted chemotherapy. To improve the therapeutic potential of aptamer-DOX conjugates, an approach that increases aptamer capacity for drug-payload without affecting aptamer target-binding is crucial. Here, we describe the use of a trifurcated Newkome-type monomer (TNM)-DOX complex to produce aptamer–TNM–DOX (Apt–TNM-DOX) conjugates. Apt–TNM-DOX can stably carry and deliver a high drug payload for eventual targeted release under a pH-controlled mechanism to inhibit lymphoma cell growth (Figure 1).

## 2. Materials and Methods

### 2.1. Chemistry Experiments

All reagents were purchased from Sigma-Aldrich (St. Louis, MO, USA) and used without further purification. Column chromatography was performed on silica gel 60 (70–230 Mesh) (Fisher Chemical, Fair Lawn, NJ, USA). Analytical thin-layer chromatography (TLC) was conducted on glass plates coated with silica gel 60 (F-254). Proton (1H) and Carbon (13C) NMR spectra were obtained on a Bruker 300 MHz magnetic resonance spectrometer (Bruker, Billerica, MA, USA) using TMS as an internal standard. NMR chemical shifts (δ) are reported in ppm. Mass spectra (ESI-MS data) were collected on an LCQ Fleet™ Ion Trap Mass Spectrometer (Thermo, Waltham, MA, USA).

### 2.2. Synthesis and Characterization of TNM-DOX

Detailed methods are described in the Appendix A.

### 2.3. HPLC Analysis

HPLC analysis was performed in a Varian 920-LC Liquid Chromatograph (Varian Inc, Walnut Creek, CA, USA) with a built-in auto-injector and UV-Vis detector. Kinetex C18 column (100 mm × 4.6 mm, 2.6 μm), (Phenomenex, Torrance, CA, USA) was used, with a flow rate of 1 mL/min; mobile phase A of 0.1% acetic acid in water; mobile phase B of acetonitrile; linear gradient of 0 to 8 min, 0–85% of B. The HPCL separation was monitored by the UV-Vis detector at 260 nm.

### 2.4. Cell Culture

CD30^+^ Karpas 299 and SU-DHL-1 cells (human anaplastic large cell line) were a gift from Dr. Mark Raffeld at NIH and HDLM2 and KMH2 cells (human Hodgkin lymphoma cell lines) were from Dr. Barbara Savoldo, Baylor College of Medicine, Houston, TX, USA. CD30^−^ control Mino cells (human B cell lymphoma cell), U937 cells (human histolytic lymphoma cell line), HL-60 cells (Human promyelocytic leukemia cell), and HEL cells (human acute myeloid leukemia cell line) were purchased from the American Type Culture Collection (ATCC, Manassas, VA, USA) and tested negative for mycoplasma. All cells used in this study were cultured in RPMI 1640 medium (HyClone, GE Healthcare Life Sciences, South Logan, UT, USA) with 10% FBS (Atlanta Biologicals, Inc., Flowery Branch, GA, USA) and 100 units/mL penicillin and 100 μg/mL streptomycin (Gibco, Waltham, MA, USA) at 37 °C under 5% CO_2_ and ≥95% humidity.

### 2.5. DOX and TNM-DOX Loading to Aptamer

To measure the affinities of DOX and TNM-3Boc to aptamer, DOX was mixed with the aptamer in molar ratios of 15:1, 30:1, and 45:1, respectively, and TNM-3Boc was mixed with aptamer in molar ratios of 6:1, 12:1, and 18:1, respectively, in 100 μL of PBS buffer for 30 min.

### 2.6. Detection of Apt–TNM-DOX Free DOX Release

DOX release from Apt–TNM-DOX was measured in solutions with serial increments of varying pH. DOX, TNM-DOX, or Apt–TNM-DOX were separately diluted with PBS of various pH (ranging from 2.0 to 8.0) to a final volume of 100 μL and the equivalent DOX concentration of 5 μM in 96-well plates. The PBS-only groups with the corresponding pH were set as blank controls. Changes in fluorescence intensity were measured using the Bio-Tek Synergic H4 microplate reader (Winooski, VT, USA).

DOX release in various pH solutions was measured using HPLC analysis. In detail, a 3 mg/mL stock solution of Apt–TNM-DOX conjugate in methanol was prepared. A 15 μL aliquot of stock solution was added into 75 μL of a series of buffered solutions with pH ranging from 2.5 to 7.4, respectively. The mixtures were incubated at RT with gentle shaking. The release of DOX was monitored with HPLC for 1 h, 2 h, 4 h, 6 h, and 24 h after incubation.

### 2.7. Stability Validation of TNM-DOX and Apt-TNM-DOX

DOX, TNM-DOX, or Apt–TNM-DOX were separately diluted with either RPMI-1640 medium containing 10% FBS or 100% human serum to a final volume of 100 μL or the equivalent DOX concentration of 5 μM in a 96-well plate. RPMI-1640 medium containing 10% FBS and 100% human serum alone were set as the blank control. All mixtures were incubated at RT for up to 24 h. The fluorescence intensity of each sample was measured at 0, 1, 2, 3, 4, 5, and 24 h using the Synergic H4 microplate reader.

### 2.8. Fluorescent Microscopy

Binding and DOX delivery of Apt–TNM-DOX were visualized using fluorescent microscopy. To this end, four CD30^+^ cell lines (Karpas299, SU-DHL-1, HDLM2, and KMH2) and four CD30^−^ control cell lines (U937, Mino, HEL, and HL-60) were used for specific binding tests. Cells were incubated with Cy5-labeled Apt–TNM-DOX at a final concentration of 100 nM in a 96-well plate at 37 °C for 60 min. To test the cell entry capacity of TNM-DOX, cells were incubated with either 2 μM of DOX or 0.667 μM of TNM-DOX at 37 °C for 30 min. To assay cell uptake, Apt–TNM-DOX containing 2 μM final concentration of DOX was incubated with Karpas 299 and U937 cells at 37 °C for 60 min. Before the end of the incubation, Hoechst 33342 (Invitrogen, Carlsbad, CA, USA) was added into cells at 37 °C for 10 min at a final concentration of 1 μg/mL. All fluorescent signals were observed using an Olympus IX81 fluorescent microscope (Olympus, Tokyo, Japan).

### 2.9. Confocal Microscopy

To detect target cell uptake of Apt-TNM-DOX, Karpas 299 and U937 cells were incubated with Cy5-labeled Apt–TNM-DOX (100 nM) and Lyso-Tracker Green DND-26 (Invitrogen) according to the manufacturer’s instructions at 37 °C for 60 min. The cells were washed once with PBS and fixed with 4% formaldehyde at RT for 10 min. After being washed with PBS, the cells were suspended in PBS and loaded onto a glass slide with a coverslip. Intracellular fluorescent signal was examined using a Nikon A1 Confocal Imaging System (Nikon, Tokyo, Japan).

### 2.10. Cell Proliferation Inhibition Assay

Karpas 299 and U937 cells were seeded in 96-well plates at densities of 7 × 10^3^ cells/well and 1 × 10^4^ cells/well, respectively. Cells were either incubated with 2 μM of TNM, Aptamer, DOX, 0.667 μM of TNM-DOX (DOX = 2 μM), Apt-DOX mixture (DOX = 2 μM), or 0.13 μM of Apt–TNM-DOX (DOX = 2 μM) at 37 °C for 10 min. Cells were then washed with PBS twice and cultured in 100 μL of fresh complete cell culture medium. Untreated cells were treated as controls and cell culture medium-only wells were set as the blank control. MTT assay was performed at one, two, and three days after drug treatment according to the manufacturer’s instructions. The relative cell proliferation rate is represented by the detected absorbance at optical density (OD) 570 nm in each specimen, using a reference wavelength of 630 nm. Cell viability = (OD of the experimental well − OD of the blank control well)/(OD of the control well − OD of the blank control well) × 100%. MTT assays were performed on CD30^+^ KMH2 and HDLM2 cells and CD30^−^ HEL and Mino cells.

### 2.11. Data Analysis

Statistical analysis and graphics presentations were carried out using Microsoft Excel software. Unpaired Student *t*-test was used in the data analysis, and *p* < 0.05 was considered a statistically significant difference.

## 3. Results

### 3.1. TNM-DOX Synthesis

The synthesis procedure for TNM (compound 1) is shown in Figure 2A and Appendix A. In brief, we synthesized compound 2 by coupling Tris base with tert-butyl acrylate as previously described [15]. Next, the amine group in compound 2 was protected with Fmoc-Cl to form compound 3 [16], and the tert-butyl group in compound 3 was removed to generate compound 4 [17]. The intermediate 4 was reacted with tert-Butyl carbazate to yield compound 5. Then, the Fmoc in 5 was removed with piperidine to produce compound 6. The protected group Boc in compound 6 was removed with TFA, then directly coupled with DOX to yield the final product 1 as previously described [18]. TNM-DOX synthesis was confirmed using mass spectrometry (Figure 2B).

### 3.2. DOX Release from TNM-DOX Is pH-Dependent

To examine the controlled payload release capacity, TNM-DOX was tested under a low pH condition, which mimics the acidic environment within tumor cell lysosomes. Free DOX drug release was monitored by HPLC analysis. A single peak in TNM-DOX was detected in the control experiment under physiological conditions (pH 7.4) (Figure 3A). After incubating TNM-DOX in acid buffer solution (pH = 5) at room temperature (RT) for 1 h, two new peaks were detected by HPLC, representing released free DOX and separated TNM structures (Figure 3B).

For further validation of pH-controlled drug release, TNM-DOX was incubated at RT in solutions under different pH conditions and free DOX release was assessed by kinetically monitoring changes in fluorescent signals derived from DOX. TNM-DOX was stable under physiological conditions (pH 7.4 and pH 6.4) with a low background level of free DOX (10–15%) after incubation for 24 h (Figure 3C). Under mildly acidic conditions (pH 5.5), ~40% of free DOX was released post 24 h incubation. Importantly, under pH 4.6–5.0 conditions, the release of free DOX reached 75–80% of the total drug payload. In the control experiment, rapid release of free DOX was detected in 1 h under pH 2.5. These results indicate that drug release of TNM-DOX is pH-sensitive and can be triggered at pH 4.6–5.5 pH, a range consistent with that of the tumor cell lysosome microenvironment [19].

### 3.3. TNM-DOX Self-Loads onto Aptamer Sequences

To assess drug delivery targeting, we attempted to covalently conjugate TNM-DOX to aptamer sequences [5]. Unexpectedly, conjugation failed when TNM-DOX bound to aptamer sequences through a strong non-covalent force. To understand whether this non-covalent conjugation of TNM-DOX resulted from DOX intercalation into aptamer sequences or TNM structure–aptamer interactions, we examined the binding patterns of aptamers with free DOX or TNM-DOX. Free DOX was mixed with aptamer sequences at different ratios and the resultant products were characterized by HPLC (Figure 4A). The peaks of aptamer sequences and free DOX were detected at 3.5 min and 4.8 min, respectively. Notably, the aptamer signal decreased as free DOX concentrations increased and a wide and flat peak with a longer retention time appeared, representing the aptamer-DOX complex. Similarly, aptamers were mixed with TNM structures, and the resultant products were analyzed by HPLC (Figure 4B). An aptamer peak was detected at 3.5 min. After mixing with the TNM structure, a new peak formed at 5.2 min, representing the TNM structure-aptamer complex. Thus, TNM structures conjugated with aptamer sequences in a non-covalent manner, revealing a new approach that allows for the self-loading of a drug into aptamers.

To validate the formation of aptamer drug conjugates, TNM-DOX was mixed with aptamer sequences, and the resultant products were analyzed by HPLC. An aptamer peak was detected at 3.5 min of retention time, a TNM-DOX peak was detected at 4.1 min, and a distinct new peak representing the aptamer–TNM–DOX conjugates appeared at 5.1 min (Apt-TNM-DOX) (Figure 4C). Notably, no free aptamer or TNM-DOX remained at a 5:1 ratio of TNM-DOX to aptamer, and approximately half of the TNM-DOX was left in the reaction at a 10:1 ratio of TNM-DOX to aptamer (Figure 4C,D). Thus, a single aptamer sequence can fully incorporate five TNM-DOX through a non-covalent self-loading approach and Apt–TNM-DOX can stably carry a high drug payload of 15 DOX molecules per aptamer sequence.

### 3.4. Controlled Drug Release by Apt-TNM-DOX

In a stability study, Apt–TNM-DOX was incubated in cell culture medium with 10% fetal bovine serum (FBS) or human serum for 24 h and free DOX release from Apt–TNM-DOX was kinetically monitored by detecting fluorescent signals of free DOX in reactions. Figure 5A,B shows that Apt–TNM-DOX was stable up to 24 h with minimal drug release. To validate controlled drug release, Apt–TNM-DOX was incubated in a medium with different pH conditions for 24 h, and changes in fluorescent signals derived from free DOX in reactions were monitored. Apt–TNM-DOX was stable when pH ≥ 6.0 (including physiological pH 7.4), released its drug payload when pH was below 6.0, and reached maximal release at pH 4.0 (Figure 5C). Thus, Apt–TNM-DOX can stably carry and deliver its drug payload under physiological conditions and efficiently release drug payload once triggered by a low pH, which characterizes tumor cell lysosomes.

### 3.5. Apt–TNM-DOX Tumor Cell Targeting and Internalization

To evaluate targeted drug delivery potential, cultured lymphoma/leukemia cells were treated for 60 min with Apt–TNM-DOX, which was labeled with Cy5 fluorescent reporter for tracking purposes. Apt–TNM-DOX specifically targeted CD30^+^ lymphoma cells (Karpas 299, SU-DHL-1, HDLM2, and KMH2), but did not react to off-target (CD30^−^) control cells (U937, Mino, HEL, and HL-60) (Figure 6A). To assess intracellular delivery capacity, cultured cells were treated with free DOX, TNM-DOX, or Apt-TNM-DOX, and intracellular signals were examined using a fluorescent microscope to detect DOX fluorescent signals. As expected, free DOX uptake occurred indiscriminately (Figure 6B) and TNM-DOX was not taken up by cells (Figure 6C). In contrast, Apt–TNM-DOX targeted lymphoma cells (Karpas 299) and delivered DOX via aptamer-mediated internalization. Intracellular delivery and Apt–TNM-DOX drug payload release were confirmed by confocal imaging of free DOX signal in lymphoma cells. No signal was detected in off-target control cells (U937) (Figure 6D). To test whether Apt–TNM-DOX signals colocalized with lysosomes, cultured cells were treated with Cy5-labeled Apt–TNM-DOX and cell lysosomes were stained with an FITC-labeled lysosome-tracker. Fluorescent and confocal imaging revealed the colocalization of Apt–TNM-DOX with lysosomes in Karpas 299 cells but not U937 cells (Figure 6E). Thus, Apt–TNM-DOX exhibits cell-specificity and intracellularly releases drug payload within lysosomes.

### 3.6. Apt–TNM-DOX Cell Specificity

To evaluate the clinical utility of Apt–TNM-DOX for targeted chemotherapy, cultured tumor cells were treated with Apt–TNM-DOX or control treatments for 10 min at 37 °C. To eliminate non-specific treatment effects, cells were washed and then cultured in fresh medium for three days. Changes in cell growth were monitored using MTT cell proliferation assays. Under aptamer guidance, Apt–TNM-DOX specifically bound to lymphoma cells, which triggered intracellular uptake and subsequent drug payload release, as shown in Figure 6. Apt–TNM-DOX treatment significantly inhibited the growth of Karpas 299 lymphoma cells but had no effect on U937 off-target control cells (Figure 7A,C). TNM-DOX treatment had no effect on cell growth as it lacked aptamer-mediated targeted delivery. In control experiments, DOX alone or an aptamer–DOX complex containing the same molar concentrations of DOX drug carried by Apt–TNM-DOX were used. Under identical treatment conditions, DOX alone or the aptamer–DOX complex had a minimal effect on cell growth. Moreover, treatments with TNM structure alone or aptamer alone did not affect cell growth. These findings confirm the potential utility of Apt–TNM-DOX for tumor cell-targeted chemotherapy. Similar inhibitory effects were observed in additional lymphoma cell lines (KMH2 and HDLM2) (Figure 7B), but not in off-target control cells (Figure 7D).

## 4. Discussion

DOX is a potent and widely used antineoplastic drug, prescribed alone or in combination with other agents to treat many carcinomas, sarcomas, and hematological cancers [20]. However, chemotherapy that consists of DOX can have side effects, including pain, nausea, vomiting, low blood count, and hair loss. As most of these side effects result from DOX-mediated toxicity in the heart, brain, liver, or kidney [20], targeted drug delivery systems are a promising therapeutic strategy. Oligonucleotide aptamers are a powerful tool to facilitate targeted drug delivery because they can act as ligands [21]. Aptamer-DOX conjugates can be easily formulated, as DOX readily intercalates into the 5′-GC-3′ or 5′-CG-3′ of double-stranded DNA or RNA. Notably, an RNA aptamer targeting prostate-specific membrane antigen was loaded with DOX at a ratio of about one DOX molecule per aptamer conjugate and then used to treat LNCaP prostate epithelial cells [22]. Another study loaded DOX onto an immature laminin receptor protein-specific DNA aptamer with a drug loading capacity of four DOX molecules per aptamer conjugate to target acute myeloid leukemia cells [23]. Likewise, Bavi et al. loaded DOX onto an annexin A1-targeted DNA aptamer with a loading capacity of 2.5 DOX per aptamer conjugate [24], and Macdonald et al. loaded DOX onto an epithelial cell adhesion molecule (EpCAM)-specific DNA aptamer with a loading capacity of 2.5 DOX per aptamer conjugate [25].

These studies demonstrate the simplicity with which aptamer-DOX conjugates can be formed and their clinical potential, but this approach can be significantly improved upon by (i) increasing the drug payload per aptamer to enhance therapeutic efficacy and (ii) introducing a controlled drug release mechanism to eliminate off-target toxicity in normal cells. Porciani et al. used nanotechnology to elongate a transferrin-specific RNA aptamer with a short DNA tail (CGA)_7_ at the 3′ end to create a duplex-G-C-rich region upon hybridization with its complementary DNA sequence (TCG)_7_, which increased DOX-loading capacity to 7.5 DOX molecules per aptamer conjugate [26]. Liu et al. designed a protein tyrosine kinase 7-specific DNA aptamer-guided tetrahedron DNA with a DOX loading capacity of 20 DOX molecules per aptamer conjugate [27]. Finally, we designed a self-assembling DNA nanoparticle that increased DOX-loading capacity to 10 DOX molecules per nanoparticle [8]. These approaches significantly increased the DOX payload per aptamer complex and the therapeutic potential of the aptamer conjugates.

To improve the therapeutic potential of aptamer-DOX conjugates, we synthesized a unique TNM-DOX complex by conjugating each TNM structure with three DOX molecules via a pH-sensitive linker. TNM structures have a high binding affinity to DNA, although the molecular mechanism underlying this non-covalent reaction is unknown [28,29]. Hydrogen bonds between TNM and oligonucleotide bases are potential binding forces between TNM-DOX and aptamer sequences (Figure 1); simply mixing TNM-DOX with aptamer sequences led to the self-loading of high drug payload into aptamers. Indeed, a TNM-DOX to aptamer ratio of 5:1 was sufficient to allow for the stable loading of 15 DOX molecules per aptamer sequence (Figure 4). Importantly, Apt–TNM-DOX was pH-sensitive and rapidly released drug payload when pH was reduced from a physiological condition (pH 7.4) to an acidic condition (pH 5.0) (Figure 5C), consistent with the microenvironment of tumor cell lysosomes [17]. Apt–TNM-DOX specifically targeted lymphoma cells, was internalized by target cells, and released a free DOX payload within cell lysosomes (Figure 6) to exclusively inhibit lymphoma cell growth (Figure 7).

Our findings provide a new approach with which to develop aptamer-based therapeutics that (1) carry a high drug payload and (2) have a pH-controlled drug release mechanism. In addition, the Apt–TNM-DOX platform can also be used for targeted chemotherapy to treat different types of cancer by using aptamer sequences specific to cells of interest.

## Figures and Tables

**Figure 1 pharmaceutics-13-01221-f001:**
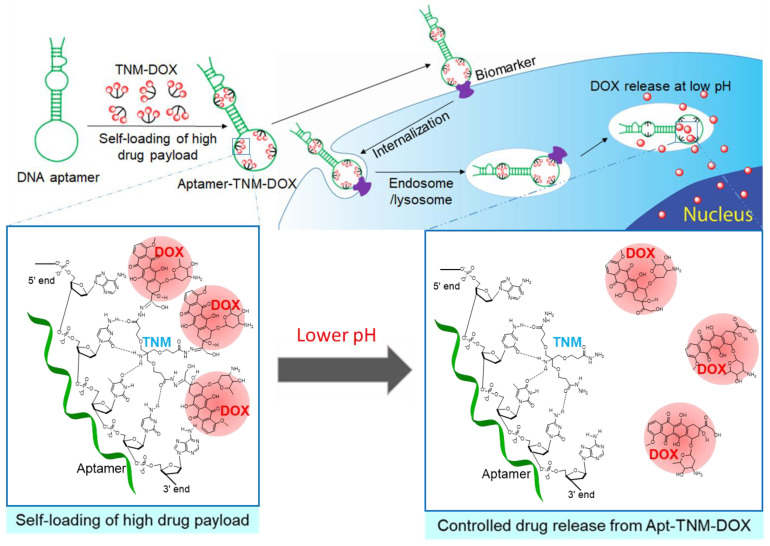
An aptamer-trifurcated Newkome-type monomer–doxorubicin (Apt–TNM-DOX) complex that carries high drug payload through a simple self-loading process and releases drug payload under a pH-controlled mechanism within cell lysosomes for targeted chemotherapy.

**Figure 2 pharmaceutics-13-01221-f002:**
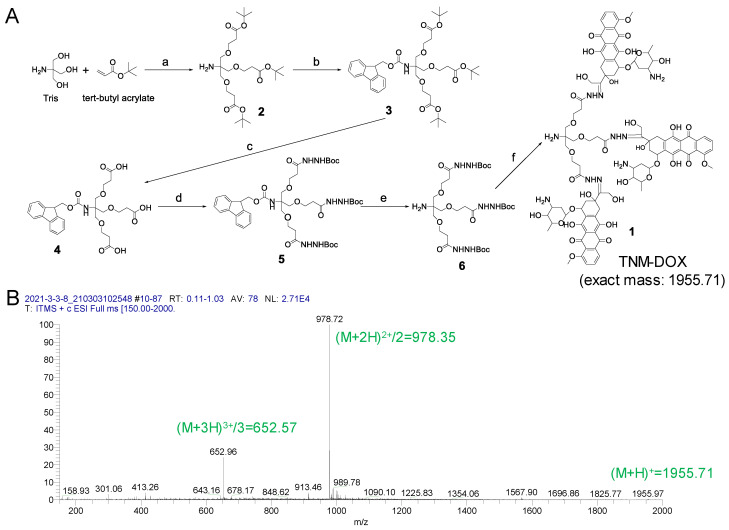
Synthesis of TNM-DOX. (**A**) The scheme of TNM-DOX synthesis. Reagents and conditions: (a) DMSO, 5 M NaOH, RT, 24 h; (b) Fmoc-Cl, DCM, NMM, RT, 1 h; (c) TFA/water = 95/5, RT, 90 min; (d) HBTU, DIEPA, DCM, tert-butyl carbazate, RT, overnight; (e) piperidine/DCM, RT, 1 h; (f) (1) TFA/DCM (1/1), RT, 30 min; (2) MeOH/AcOH/Pyridine (10/0.1/0.1), DOX, RT, overnight. (**B**) Mass Spectrum of TNM-DOX. MS (ESI): [M + H]^+^ = 1955.7, [M + 2H]^2+^/2 = 978.4, [M + 3H]^3+^/3 = 652.6. (DMSO = Dimethyl sulfoxide, Fmoc-Cl = Fluorenylmethyloxycarbonyl chloride, DCM = Dichloromethane, NMM = N-Methylmorpholine, RT = room temperature, TFA = Trifluoroacetic acid, HBTU = N,N,N′,N′-Tetramethyl-O-(1H-benzotriazol-1-yl) uronium hexafluorophosphate, DIPEA = N,N-Diisopropylethylamine, MeOH = Methanol, AcOH = Acetic acid, DOX = Doxorubicin).

**Figure 3 pharmaceutics-13-01221-f003:**
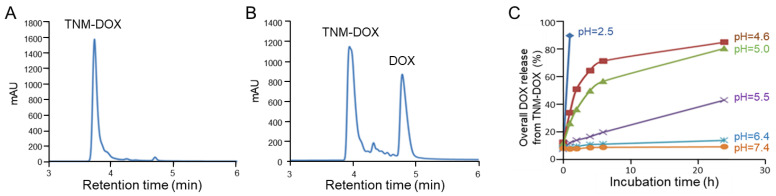
Free DOX release from TNM-DOX. (**A**) HPLC analysis of synthetic TNM-DOX. (**B**) HPLC analysis of free DOX release (in milli absorbance units, mAU) from TNM-DOX over time. (**C**) Time course of free DOX release from TNM-DOX under different pH conditions. Release of free DOX at 1, 2, 4, 6, and 24 h post-incubation in buffers at different pH as indicated.

**Figure 4 pharmaceutics-13-01221-f004:**
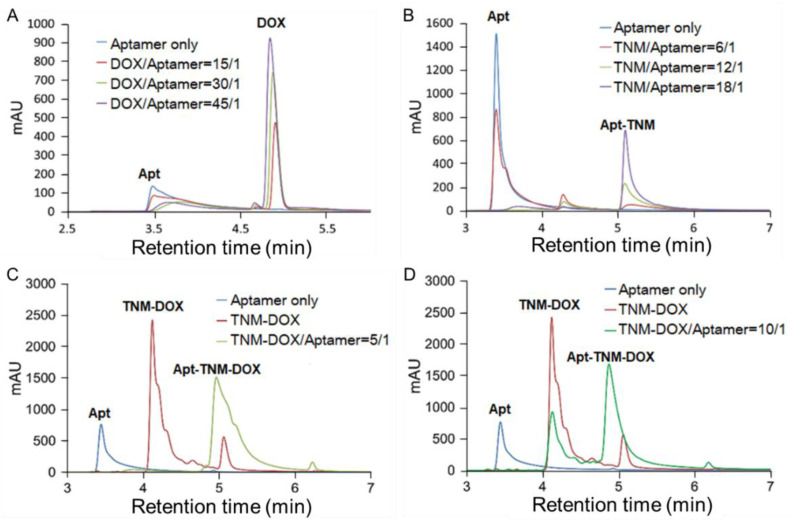
TNM structure mediates self-loading of a high drug payload into aptamer sequences. (**A**) HPLC analysis of the aptamer-DOX complex. (**B**) HPLC analysis of TNM structure conjugated with aptamer sequences. (**C**,**D**) HPLC analysis of aptamer–TNM–DOX conjugate formation at molar ratios 5:1 and 10:1 of TNM-DOX to aptamer, respectively. Aptamer–TNM–DOX conjugates carried a high drug payload of 15 DOX molecules per aptamer sequence.

**Figure 5 pharmaceutics-13-01221-f005:**
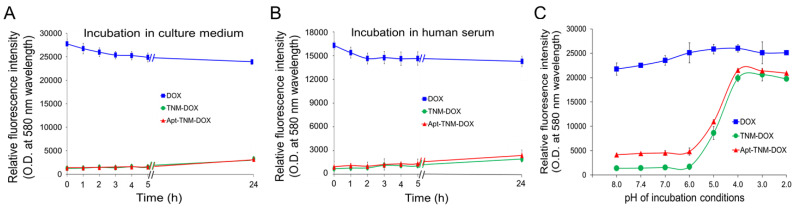
pH-controlled drug release by Apt-TNM-DOX. (A and B) Stability assays. TNM-DOX and Apt–TNM-DOX were incubated in culture medium (**A**) or human serum (**B**). Changes in fluorescent signals from free DOX were measured kinetically. (**C**) Drug release assay. TNM-DOX and Apt–TNM-DOX were incubated in culture medium at different pH conditions as indicated at 37 °C for 30 min. Resultant changes in fluorescent signals from released free DOX under each condition were quantified.

**Figure 6 pharmaceutics-13-01221-f006:**
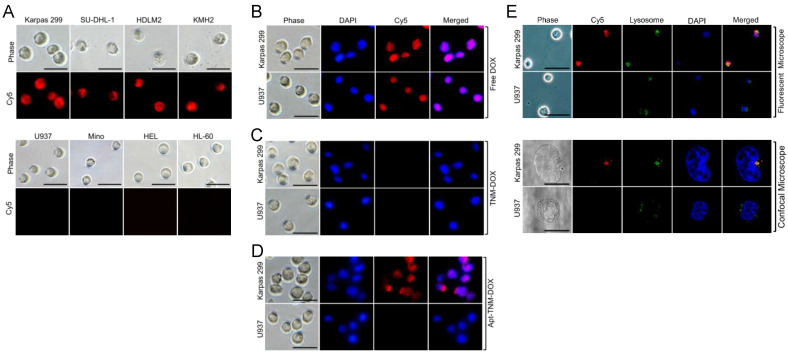
Cell-specific delivery and intracellular release of drug payload by Apt-TNM-DOX. (**A**) Cy5-labeled Apt–TNM-DOX specifically bound to CD30^+^ lymphoma cells (Karpas 299, SU-DHL-1, HDLM2, and KMH2), but did not react to off-target CD30^−^ control cells (U937, Mino, HEL, and HL-60). (**B**) DOX drug freely penetrated tumor cells without specificity detected by DOX fluorescent signals. (**C**) TNM-DOX did not show cell uptake. (**D**) Intracellular delivery and release of DOX payload by non-labeled Apt–TNM-DOX occurred in Karpas 299 lymphoma cells but not in off-target U937 control cells. (**E**) Colocalization of Cy5-labeled Apt–TNM-DOX and cellular lysosomes stained by FITC-lysosome tracker was confirmed in Karpas 299 cells but not in off-target U937 cells by fluorescent microscopy (upper panels) and confocal microscopy (lower panels). Cell nuclei were stained with DAPI fluorescent dye in B-E. Scale bars are 10 μm in **E**, and 25 μm in others.

**Figure 7 pharmaceutics-13-01221-f007:**
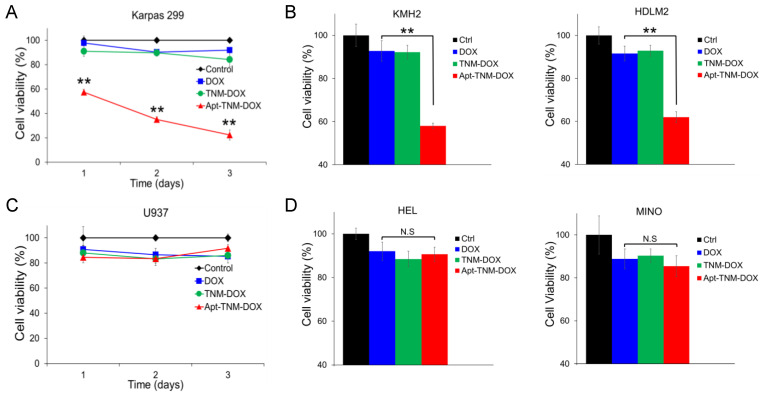
Apt–TNM-DOX specifically targets lymphoma cells and inhibits cell growth. (**A**) Apt–TNM-DOX treatment inhibited the growth of Karpas 299 lymphoma cells. (**B**) Three days after Apt–TNM-DOX treatment, a similar inhibitory effect was observed in KMH2 and HDLM2 lymphoma cells. (**C**) Apt–TNM-DOX treatment had no effect on off-target U937 control cells. (**D**) No inhibitory effect on additional off-target control cells, HEL and Mino cells was observed three days after Apt–TNM-DOX treatment. ** *p* < 0.01. N.S. indicates that the difference was not significant.

## Data Availability

Not applicable.

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
