# Peer review of "Aptamers with Self-Loading Drug Payload and pH-Controlled Drug Release for Targeted Chemotherapy"

_pharmaceutics, 2021, doi:10.3390/pharmaceutics13081221_

Round 1

Reviewer 1 Report

The manuscript submitted by Zeng et al. entitled "Aptamers with self-loading drug payload and pH-controlled 2 drug release for targeted chemotherapy" aims to characterize the functional activity of a new aptamer-TNM-DOX conjugate in lymphoma cells. The work reveals the new aptamer carries a high drug payload, releasing the DOX in an pH-dependent fashion with no non-toxic lateral effects. The M&M are well described and results well discussed. However, it will be very useful for the readers if the authors clarify two major points:

  • when the lymphoma cells were exposed to the aptamer, do they enter in senescence or apoptosis (the MTT assays does allow to distinguish these two scenarios). Please used Annexin V (e.g.)
  • if the duration of the treatment is only 5' or 20', the authors will obtain similar effects?

Author Response

Reviewer #1 comments

Comment 1: When the lymphoma cells were exposed to the aptamer, do they enter in senescence or apoptosis (the MTT assays does allow to distinguish these two scenarios)? Please used Annexin V (e.g.)

Response1: Thanks for reviewer’s comment. This study showed that Apt-TNM-DOX treatment inhibited lymphoma cell growth, which was most likely through cellular apoptosis pathway. This conclusion was supported by our previous study of cellular effects of aptamer and DOX on lymphoma cells. As showed in Figure 1D, Leukemia. 2016;30(4):987-991. doi:10.1038/leu.2015.216, aptamer-DOX chemotherapy induced inhibition/death of targeted cells through cellular apoptosis pathway, which was demonstrated by Annexin V method. In addition, monomer aptamer treatment alone had minimal effect on lymphoma cell death/viability (Figure 7C of Biomaterials. 2013;34(35):8909-8917. doi:10.1016/j.biomaterials.2013.07.099).

Comment 2: If the duration of the treatment is only 5' or 20', the authors will obtain similar effects?

Response2: Thanks for reviewer’s comment. To evaluate cell binding capacity of the aptamer, we performed a time-course study (unpublished). The Cy3-labeled aptamer was incubated with lymphoma cells and resultant cell binding of aptamer was detected at different time points (0-120 min) by fluorescent microscopy (Figure 1). Significant cell binding of aptamer was observed in 10 min, and cell binding reached maximum in 60 min. Theoretically, longer incubation of aptamer-TNM-DOX with target lymphoma cells (20 min vs. 5 min) will cause more cell binding and intercellular delivery of carried drug payload. To study aptamer-mediated targeted therapy in this study, cells were treated for 10 min and washed to remove unbound aptamer-TNM-DOX or free DOX. We believe that the short incubation approach will eliminate non-specific effect of free DOX that can diffuse into cells freely. Therefore, under this condition, the observed cellular effect was caused by aptamer-mediated targeted chemotherapy, not due to non-specific diffusion of free DOX drug.

Figure 1: Time-course study of aptamer binding to lymphoma cells. Cultured Karpas 299 cells were incubated with Cy3-labeld aptamer probe and resultant cell binding was detected under a fluorescent microscope at different timepoints as indicated.

Reviewer 2 Report

This is a very interesting and well written manuscript. The methods are appropriate and well described. The data are sound. The discussion and the deduced conclusions are well balanced and adequately supported by the data. The title and abstract accurately convey what has been found. The Figures are clear. The writing is acceptable.

I recommend this manuscript for publication in Pharmaceutics and suggest the authors to incorporate minor changes in accordance with the corrections and comments of the attached file.

Author Response

Reviewer #2 comments

 Comment 1: Please, uniform through all the text (probably only in subsection 2.1 and the supporting materials) how to name the various synthesized compounds: “compound #x”, “compound x”, “product #x”. With or without the symbol “#”?

Response 1: Thank you for the comment. The compounds names were renamed by removing the symbol “#”.

Comment 2: Please uniform the legends of the Figures: some legend’s titles are in bold some others not; some panels’ name are in bold some others not [(A), (B), …(A), (B),..]

Figure 2. Synthesis of TNM-DOX. ………

Figure 3. Free DOX release from TNM-DOX………….

Figure 4. TNM structure mediates self-loading of a high drug payload into aptamer sequences………..

Figure 5. pH-controlled drug release by Apt-TNM-DOX……..

Figure 6. Cell-specific delivery and intracellular release of drug payload by Apt-TNM-DOX………..

Figure 7. Apt-TNM-DOX specifically targets lymphoma cells and inhibits cell growth…………..

Response 2: Thanks for the comment. The legends of the Figures were uniformed.

Comment 3: In Figure 1: “Aptamer sequence” Should be “DNA aptamer”. “Low pH condition”

Should be “Lower pH”

Response 3: Thanks for the recommendation. The Figure 1 was revised.

Comment 4:  In the legend of Figure 2, the evidenced “+”, “2+” and “3+” (see below) should be superscript; last sentence should be removed; “DIEPA” should be “DIPEA” (also in the supporting materials):

Figure 2. Synthesis of TNM-DOX. (A) The scheme of TNM-DOX synthesis. Reagents and conditions: a) DMSO, 5 M NaOH, RT, 24 h; b) Fmoc-Cl, DCM, NMM, RT, 1 h; c) TFA/water = 95/5, RT, 90 min; d) HBTU, DIEPA, DCM, tert-butyl carbazate, RT, overnight; e) piperidine/DCM, RT, 1 h; f) (1) TFA/DCM (1/1), RT, 30 min; (2) MeOH/AcOH/Pyridine (10/0.1/0.1), DOX, RT, overnight. (B) Mass Spectrum of TNM-DOX. MS (ESI): [M+H]+ =

Response 4: Thanks for your comments. The errors were corrected.

Comment 5:  In Figure 3, panel C, y axis title: “Cumulative DOX release…” Should be “Overall DOX release…”

Response 5: Thanks for the comment. The Figure 3 was revised.

Comment 6:  Pag 8, line 245: seqeunce (Figure 4) should be sequence (Figure 4)

Response 6: Thanks for the comment. The typo was corrected.

Round 2

Reviewer 1 Report

The manuscript is suitable for publication in the present form since the authors positively answered all the questions raised by this reviewer.